

# Application of multimodal perception scenario construction based on IoT technology in university music teaching

Yuexia Gao

Music College, Hubei Normal University, Huangshi, Hubei, China

## ABSTRACT

In the contemporary landscape of diversified talent cultivation, enhancing education through intelligent means and expediting the process of talent development stand as paramount pursuits. Within the domain of instrumental music education, beyond merely listening to student performances, it becomes imperative to assess their movements, thus furnishing additional insights to fuel their subsequent growth. This article introduces a novel multimodal information fusion evaluation approach, combining sound information and movement data to address the challenge of evaluating students' learning status in college music instruction. The proposed framework leverages Internet of Things (IoT) technology, utilizing strategically positioned microphones and cameras within the local area network to accomplish data acquisition. Sound feature extraction is accomplished through the employment of Mel-scale frequency cepstral coefficients (MFCC), while the OpenPose framework in deep learning and convolutional neural networks (CNN) is harnessed to extract action features during students' performances. Subsequently, the fusion of feature layers is achieved through CNN, culminating in the evaluation of students' academic efficacy, facilitated by a fully connected network (FCN) and an activation function. In comparison to evaluations conducted by the teacher in the class, this approach achieves an impressive accuracy of 95.7% across the three categories of Excellent, Good, and Failed students' evaluation processes. This breakthrough offers novel insights for the future of music teaching and interactive class evaluations while expanding the horizons of multimodal information fusion methods' applications.

# INTRODUCTION

Music teaching in universities is an important way to cultivate musical talents and is of great significance in maintaining and developing music culture in China. Music education is not only about imparting skills and knowledge, but also about expressing human emotions, values and cultural heritage through music, which is vital in the cultivation of students' psychological and cultural qualities (*Pardayevna, 2022*). Therefore, it is very important to improve the level and quality of music teaching in universities. Recently, with the development and popularity of IoT, multimodal sensing scenario construction based on IoT has been widely used in various fields (*Wang, 2022a*). In university music teaching,

Corresponding author
Yuexia Gao,
dongfangshenlu11@sina.com

the use of IoT can provide teachers with more comprehensive and accurate data about students' performance, help teachers better assess students' performance levels and provide personalized guidance and feedback, thus improving students' learning outcomes. Among them, sound and movement are two important aspects that affect students' performance level, so it is meaningful to conduct joint analysis of sound and movement during students' performance (*Han, 2022*).

Traditionally, music teaching is usually done in a face-to-face manner, *i.e.,* students learn music theory and performance techniques in class with the teacher, and the teacher assesses and instructs students' performance level by listening, watching and guiding. However, there are some problems with the traditional face-to-face teaching method, such as teachers can hardly assess students' performance level comprehensively and accurately because they can only judge by their own sensory experience; moreover, each student's learning progress and needs are different, so it is difficult for teachers to provide individual guidance and feedback to each student. Therefore, multimodal perceptual scenario construction based on IoT can make up for the shortcomings of traditional music teaching and improve students' learning outcomes, while providing teachers with more comprehensive and accurate data on students' performance and providing students with more personalized and effective learning guidance (*Fernández-Barros, Duran & Viladot, 2023*).

The purpose of this article is to introduce a multimodal perceptual scenario construction method based on IoT, which enables the assessment and guidance of students' performance level by jointly analyzing the sounds and actions during their performance. Specifically, we will use a multimodal fusion approach in deep learning, using a convolutional neural network for feature extraction of audio and OpenPose for feature extraction of action, fusing the features of the two DNNs together, and finally feeding the fused features into a fully connected neural network (FCN) for classification or regression tasks (*Wang, 2022b*). The contributions of this article are as follows:

1. A music learning evaluation framework based on multimodal fusion of performance sound data and student movement data of students is established to evaluate their performance in college music teaching;

2. Extracted sound signal features and human motion features using the MFCC method as well as the OpenPose framework;

3. The objective evaluation of the three types of student performance was completed by fusing the characteristics of sound signals and motion information, and the results agreed with the expert teachers' evaluation results by 95.7%.

The rest of this article is arranged as follows: In 'Related works', we introduce related works for IoT and the intelligent application in music teaching; In 'Music performance evaluation based on performance sound characteristics and movement characteristics', the MFCC and OpenPose are described to extract the feature; 'Experiment result and analysis' gives the experiment and result analysis of the students' performance evaluation; In 'Discussion', we discuss the result and advice for the music education; the 'Conclusion', is presented at the end.

## RELATED WORKS

### Research on the application of IoT and artificial intelligence in music teaching

With the development of IoT, the application in the music education has become increasingly common. Many researchers have used these technologies to develop intelligent music teaching systems to improve students' learning and interest. For example, *Inoue & Otsuka (2019)* proposed an intelligent music education system that uses IoT technology and virtual orchestra technology to collect and analyze students' performance data and provide real-time feedback (*Wei, Karuppiah & Prathik, 2022*). *Chen, Lin & Yen (2019)* personalized music learning instruction for students using multimodal physiological signal analysis technology. *Nguyen et al. (2021)* developed an intelligent music education system based on machine learning algorithms that automatically analyzes students' performance and provides feedback and suggestions accordingly. These studies suggest that artificial intelligence and IoT technologies can bring many new opportunities and challenges to music education. *Zhang et al. (2021)* introduce an intelligent music teaching system based on IoT and cloud computing that can provide a more convenient and efficient learning environment for students. *Zhou, Liu & Wang (2021)* introduced design and implementation process of a smart music education system using IoT that can provide personalized music learning guidance. *Zou et al. (2021)* proposed a music education support system developed using machine learning algorithms that can automatically adjust the difficulty level based on students' performance and provide feedback and suggestions accordingly. *Goh, Tien & Chen (2021)* introduced a novel music education system based on multi-level knowledge mining technology that can analyze students' performance from multiple dimensions and provide appropriate learning guidance and suggestions.

The above study reveals that with the improvement of various data collection methods in IoT technology, and the application research, its application research in music education is also improving. The music education is improved through the integration of different signals.

Although these studies demonstrate the extensive utilization of IoT technology in music education, they still encounter certain technical constraints. Firstly, the implementation of IoT devices and sensors to gather student performance data may be confined by the precision and efficiency of the devices, leading to inadequate and imprecise data collection. Secondly, the utilization of AI algorithms for data analysis and student evaluation may necessitate substantial computational resources and processing capabilities, posing challenges, particularly in large-scale music education scenarios. Additionally, compatibility issues between various IoT devices and platforms may augment the complexity in system development and deployment. Consequently, as we further advance the development of intelligent music education systems, these technical limitations necessitate perpetual attention to ensure the system's reliability, stability, and scalability, thereby providing an enhanced music education experience for students.

**Research on human behavior evaluation based on multimodal data**

Multimodal fusion techniques can fuse information obtained from multiple sensors to improve the accuracy and robustness of action recognition. Several studies have demonstrated that multimodal information can provide a greater amount of information and a more comprehensive view to evaluate human actions compared to a single sensor. *Su et al. (2019)* proposed a multimodal data fusion approach for human action recognition. A three-dimensional spatio-temporal tensor was constructed by fusing data from RGB video, depth images, and IMU sensors. Then, features were extracted using CNN and classified using a LSTM model. *Song et al. (2017)* used an end-to-end spatial–temporal convolutional neural network (ST-CNN) for action recognition of multimodal data. They fused RGB video and depth image data together and used ST-CNN to model both spatial and temporal features to obtain more accurate action recognition results. *Bao et al. (2019)* proposed a multimodal sensor-based deep neural network approach to evaluate surgical skills. By capturing motion and movement using visual and inertial sensors, a deep neural network combined with temporal information is used to classify surgical skills. This study shows that multimodal fusion can improve the accuracy of surgical skill assessment. *Wu et al. (2018)* proposed a method for joint learning of human action recognition and pose estimation under multiple perspectives. The method utilizes a CNN to extract visual features and RNN and to implement spatio-temporal feature modeling. This study shows that multimodal fusion can improve the accuracy of action recognition and pose estimation. *Mavroudi et al. (2018)* proposed a multimodal sensor-based method for upper limb motor function rehabilitation. The method utilizes visual, electrical signals and inertial sensors for motion and movement capture and evaluation, and rehabilitation of upper limb motor function through joint analysis of multimodal data. This study showed that multimodal fusion can improve the effectiveness of the rehabilitation process. *Liu et al. (2019)* proposed a multimodal fusion action recognition method based on deep convolutional neural networks. The method classifies actions by combining visual, acoustic, and sensor data, and uses a multi-branch deep convolutional neural network for feature extraction and fusion. This study shows that multimodal fusion can improve the action recognition performance. *Andrade-Ambriz et al. (2022)* used a deep learning model to fuse the two sensor data to improve the accuracy of action recognition using video and IMU data. *Dai, Liu & Lai, (2020)* proposed a multimodal sensor fusion method based on a Bi-LSTM model for human activity recognition. The authors fused accelerometer and gyroscope data to achieve more accurate and robust human activity recognition.

The primary focus of music education lies in the precision of intonation and the refinement of students' movements during the process of playing or singing. Therefore, integrating sound data and human movements to evaluate the overall performance course is a judicious approach. In this article, we opt for a deep learning model that incorporates multimodal information to analyze the progress of music education. Through this model, we conduct a comprehensive examination of both the students' movements and their performance of the repertoire in unison.

# MUSIC PERFORMANCE EVALUATION BASED ON PERFORMANCE SOUND CHARACTERISTICS AND MOVEMENT CHARACTERISTICS

## Sound information and movement information collection based on IoT technology

With the continuous development of IoT technologies, more devices and sensors can be connected to the Internet for information transmission and sharing (*Ding, Tukker & Ward, 2023*). Among them, sound information and video motion information collection based on IoT technology has become one of the hot spots of current research. By using microphones and cameras in IoT devices, the acquisition of surrounding sounds and movements can be realized. The collected sound information and video action information can be applied to many fields, such as smart home, smart city, smart health, *etc*. For example, in smart home, the sound and action information of family members can be collected to achieve intelligent control of home devices and improve the comfort and convenience of family life.

In this article, data collection was carried out based on local area network information using a cell phone microphone and a webcam built. The network was set up in a designated room through a local routing device, and then the microphone and webcam were connected to this network and the data was collected in real time through a monitored laptop (*Lita et al., 2007*). The sound acquisition system in this article is a self-designed module, which mainly contains a microphone module, a signal conversion module and a signal processing module and a data storage module. Among them, the microphone module is used to capture sound signals, which usually includes microphone arrays, unidirectional microphones, stereo microphones, and other types. Different types of microphones are suitable for different scenarios, and the appropriate microphone can be selected for installation and configuration. The signal conversion module is used to convert the analog signal captured by the microphone to digital signal, which usually includes analog-to-digital converter (ADC) and digital-to-analog converter (DAC). The signal processing module is used to process and analyze the digital signals to extract useful information. Common signal processing algorithms include Fourier transform, filtering, noise reduction, gain control, *etc*. Finally, wireless network is used to store the data locally, thus completing the collection of sound data and video data. The established system and the whole data collection process is shown in Fig. 1. The volunteer conducted the experiment in the particular rooms and finished the test without others' presence.

## Sound feature extraction based on MFCC features and CNN

In the classification of audio signals, feature extraction is a crucial step because it can downscale the high-dimensional input signal into more representative low-dimensional features. Traditional audio feature extraction methods such as MFCC, LPC, LPCC, *etc*. These methods have solved the problem of audio signal classification to a certain extent, but since they are based on artificially designed features, there are problems such as insufficient feature representation and information loss. In this article, the collected performance

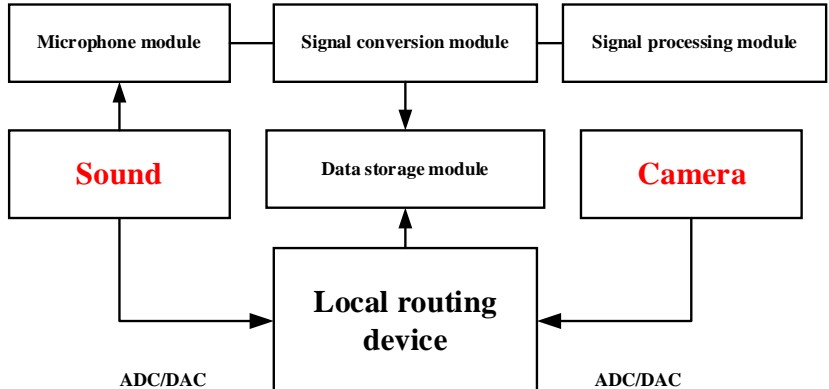

**Figure 1** The framework for the data collection.

data are firstly subjected to mel-frequency cepstral coefficients (MFCC) variations, and then CNNs are used for further feature extraction (*Deng et al., 2020*) MFCC is a feature extraction method widely used in speech signal processing, and its basic principle is to simulate the auditory properties of the human ear by converting the sound signal into a set of coefficients that can describe its spectral characteristics. Specifically, MFCC is divided into the following steps: first, it performs framing: the sound signal is divided into several frames, each of which is usually 20–30 ms in size. The purpose of this step is to convert the sound signal into a time-varying signal. Next, a window is added, and a window function is applied to each frame, usually using a Hamming or Haining window. The purpose of the window function is to suppress spectral leakage at the frame edges, making the spectral estimation more accurate. The sound signal of each frame is multiplied by the window function w(n) to obtain the window function weighted signal (*Chowdhury & Ross, 2019*).

$$y(n) = x(n)w(n) \tag{1}$$

After completing the windowing of the data amount it is necessary to perform a Fourier transform on it, which gives the frequency domain representation of the signal:

$$Y(k) = \sum_{n=0}^{N-1} y(n) e^{-j2\pi nk/N} \tag{2}$$

where k denotes the frequency and N denotes the length of the FFT. The frequency domain signal is Mayer filtered to obtain the output of a set of Mayer filter banks. The Mel filter bank serves to simulate the perceptual properties of the human ear for sound frequencies by converting the frequency axis in the spectrogram into a Mel scale.

$$H_m(k) = \begin{cases} 0, & k < f(m-1) \\ \dfrac{k - f(m-1)}{f(m) - f(m-1)}, & f(m-1) \le k \le f(m) \\ \dfrac{f(m+1) - k}{f(m+1) - f(m)}, & f(m) \le k \le f(m+1) \\ 0, & k > f(m+1) \end{cases} \tag{3}$$

**Peer**J Computer Science

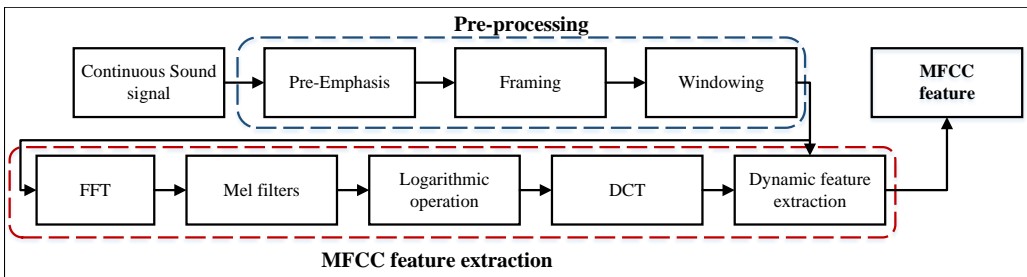

**Figure 2  The framework for the MFCC feature extraction.**

where f(m) denotes the center frequency of the first m The center frequency of the Meier filter is usually calculated using the following equation.

$$f(m) = 700 \times \left(10^{m/2595} - 1\right) \tag{4}$$

where m denotes the number of the Meier filter, which usually takes the value $0 \sim M - 1$, where M is the number of Mel filters. Finally, the output of the Mel filter set is logarithmically taken to obtain a set of Mel-frequency cepstral coefficients (MFCC). The purpose of this step is to enhance the low-frequency components and suppress the high-frequency components, so that the MFCC can better reflect the frequency perception characteristics of the human ear. The output of each Mel filter bank is Hm(k) is taken logarithmically to obtain a set of Meier frequency cepstral coefficients (MFCC).

$$C_m = \sum_{k=0}^{N-1} \log|Y(k)|^2 H_m(k) \tag{5}$$

where $|Y(k)|$ denotes the amplitude of the signal in the frequency domain and represents the energy distribution of the sound signal in the frequency domain. MFCC has been widely used in speech recognition, where the features of the sound signal can be extracted by MFCC and fed into machine learning algorithms for classification. The main advantage of MFCC is that it can simulate the auditory characteristics of the human ear and better reflect the characteristics of the sound signal, which improves the accuracy and robustness (*Meghanani, Anoop & Ramakrishnan, 2021*). The framework of this feature extraction is shown in Fig. 2.

After completing MFCC feature extraction this article uses CNN method for further feature extraction. the core idea of CNN is convolution and pooling, convolution can extract spatial or temporal correlation in image or audio, while pooling can reduce the size of feature map and improve the computational efficiency. In audio signal processing, the convolution kernel is usually 1D, because the audio signal is a 1D signal. Assuming that the input audio signal is $x$, the output can be expressed as:

$$h_i = f_i\left(w_i * h_{i-1} + b_i\right) \tag{6}$$

where $w_i$ is the weight of the convolution kernel, $h_{i-1}$ is the output of the previous layer, $b_i$ is the bias term, and $f_i$ is the activation function. For audio signals, $w_i$ is usually a 1D

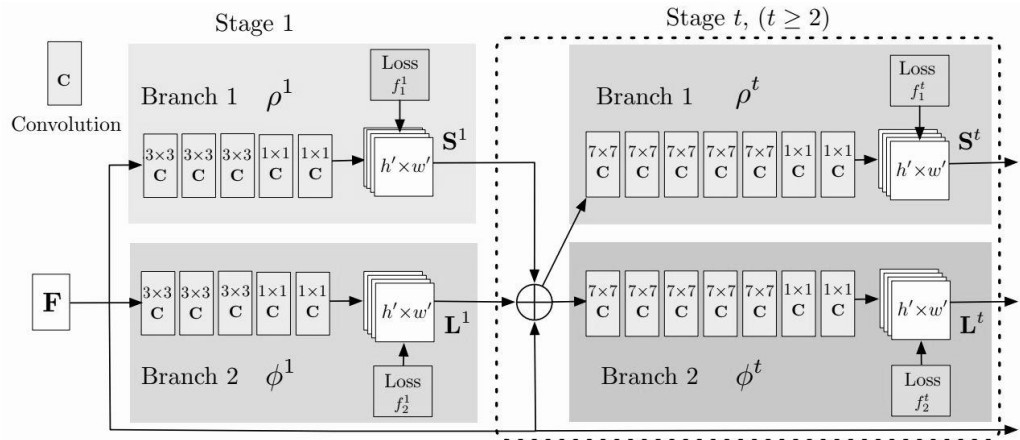

**Figure 3  The framework for the OpenPose.**

convolution kernel with a window size of *n*, where *n* is usually a fraction of the sample rate of the audio signal. Through the convolution of MFCC variations and CNN networks we obtain the sound features, based on which the motion features are further extracted using OpenPose to achieve multimodal data fusion analysis.

## Motion information extraction based on OpenPose

OpenPose is a deep learning-based human pose estimation framework designed to extract human pose information by using images or videos captured by cameras. This framework is widely used in many fields such as human behavior analysis, human–computer interaction, game development, and healthcare (*Shi et al., 2022*).

OpenPose uses convolutional neural networks (CNNs) to detect human joints and represent each joint as a coordinate. OpenPose is capable of detecting and estimating poses for multiple individuals within the same image or video frame. It can handle crowded scenes and provide pose information for each person simultaneously. OpenPose is optimized for real-time or near real-time performance on CPUs and GPUs. It efficiently processes images or video frames to provide pose estimation results in real-time, which is essential for applications like human–computer interaction, augmented reality, and gaming. Specifically, the framework uses two parallel CNNs to extract information about the body and body parts (*Lin et al., 2022*). The outputs of these two CNNs were used to generate confidence heat maps and joint offset matrices. In OpenPose, each joint is represented by a heat map and an offset matrix. These heatmaps contain the pixels associated with the joint and indicate the probability that the pixel is part of the joint. The joint offset matrix contains the exact coordinate information of the joint position. the overall network flowchart of OpenPose is shown in Fig. 3.

In Fig. 3, the model is divided into two networks St and Lt , where St is the confidence map of the nodes output at stage t and Lt is the affinity of two-two relationship nodes output at stage t, *i.e.,* the weight coefficient. Both can be expressed by Eqs. (7) and (8):

$$S^t = \rho^t(F, S^{t-1}, L^{t-1}), \forall t \geq 2 \tag{7}$$

$$L^t = \varphi^t(F, S^{t-1}, L^{t-1}), \forall t \geq 2 \tag{8}$$

The loss values of each stage can be expressed by Eqs:

$$f_s^t = \sum_{i=1}^{J} \sum W(p) \cdot \left\| S_i^t(p) - S_i^*(p) \right\|_2^2 \tag{9}$$

$$f_L^t = \sum_{j=1}^{J} \sum W(p) \cdot \left\| L_j^t(p) - L_j^*(p) \right\|_2^2 \tag{10}$$

$$f = \sum_{t=1}^{T} (f_s^t + f_L^t) \tag{11}$$

where the one with superscript $^*$ represents the true value, the one with superscript t represents the predicted value at different stages, p represents each pixel point, W (p) represents the missing mark at this point, and its value is 0 or 1. If it is 0, the loss value is calculated, as shown in the overall loss value Eq. (11). After completing the model training we use the projection relationship between two points in the limb for the calculation of the angle between each joint, as shown in Eq. (12):

$$\theta_n = \arccos \frac{r_{i,j}^1 \cdot r_{i,k}^1 + r_{i,j}^2 \cdot r_{i,k}^2}{\sqrt{r_{i,j}^{12} + r_{i,j}^{22}} + \sqrt{r_{i,k}^{12} + r_{i,k}^{22}}} \tag{12}$$

From this, we can get the fusion features based on MFCC sound signal feature extraction and student action information extraction for student learning effectiveness evaluation.

## EXPERIMENT RESULT AND ANALYSIS

In this article, to better classify the students' learning effects, we selected students of different grades and with large differences in performance in the exams to perform the fixed repertoire of the exams as required. As feature combination is very important for the model training, we just recorded the training loss of the proposed model. The experiment data recorded is the results of the classification using the combination feature. During the performance process, we collected audio data and video movement data from the students for subsequent model analysis. In the test we evaluated the model using Precision, Recall and F1 score during the analysis of the model, and the results are shown below.

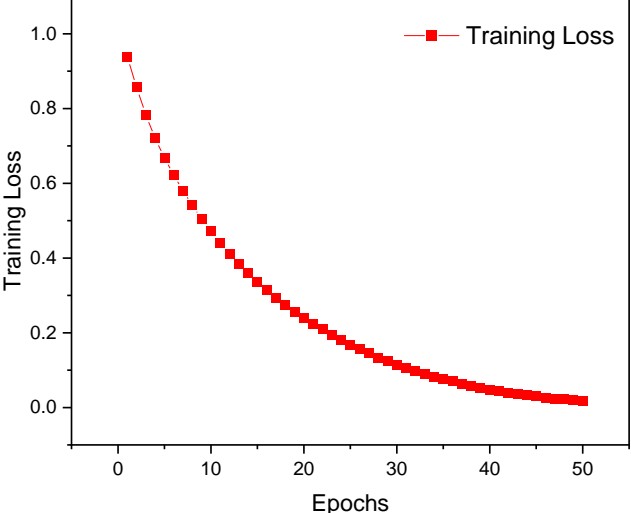

**Figure 4** **The training loss for the proposed framework.**

## Model training and student learning outcomes assessment

In this article, after completing the model construction, the logarithmic loss function is used for model training, and the overall loss value change curve is shown in Fig. 4. By observing the change of training loss we can find that the model training is more reasonable and the overall loss change is smooth, which indicates that the performance of the model is more balanced.

To analyze the relationship between the data used and the model, we give the model evaluation metrics for the training phase, validation phase and testing phase, the results of which are shown in Fig. 5.

The results depicted in Fig. 5 offer valuable insights into the performance of the model during different stages of the training process. It is evident that the accuracy achieved during the training phase is marginally superior when compared to that observed in the validation and testing stages. However, it is crucial to emphasize that the dissimilarity in accuracy levels between these stages is relatively minor. This observation indicates that the model exhibits commendable generalization capabilities, as the recognition accuracy achieved during the testing stage remains competitive with that attained during the training phase.

To gain a more comprehensive understanding of the model's performance, it is imperative to delve into the analysis of the corresponding loss function curves during the training process. These curves provide essential information about the model's optimization and learning dynamics. A thorough examination of the loss function behavior can shed light on potential overfitting issues that may compromise the model's generalization capacity.

Upon scrutinizing the loss function curves, it becomes apparent that the model converges gracefully during the training phase. This smooth convergence suggests that the model does not suffer from overfitting, as no pronounced increase in the loss is evident for the

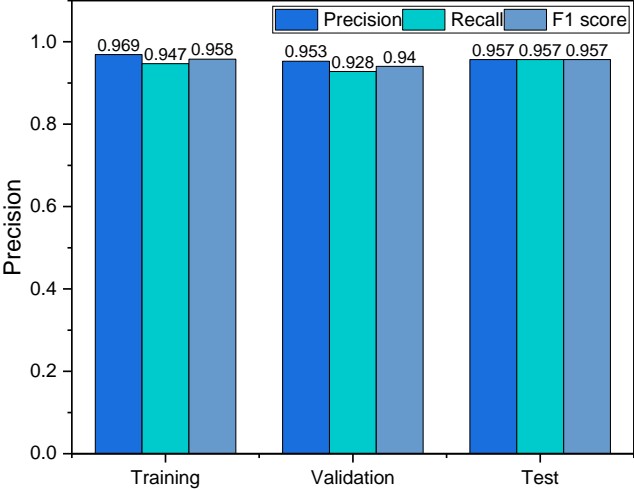

**Figure 5** The result for the student's performance evaluation.

validation or testing datasets. Consequently, the consistent accuracy levels achieved across training, validation, and testing stages underscore the robustness and generalizability of the proposed model.

It is also essential to consider the implications of the observed balanced recognition accuracy. The balanced accuracy signifies that the model does not favor any specific class or category during its predictions, resulting in a fair and unbiased performance across different classes. Such an equitable distribution of recognition accuracy is highly desirable, as it indicates that the model can effectively handle various inputs without displaying significant performance discrepancies across different classes.

Moreover, it is noteworthy that the minor discrepancy between the training accuracy and the accuracy during validation and testing stages could be attributed to several factors. For instance, variations in the data distribution between the training and validation/testing datasets, as well as the inherent stochasticity in the optimization process, may contribute to the observed differences. Nonetheless, the overall balanced accuracy and the lack of overfitting validate the efficacy and reliability of the proposed model in tackling real-world recognition tasks.

## Recognition results of learning effects under different features

To verify the recognition accuracy of the multimodal perception model proposed based on the fusion of feature layers, we have tested the recognition accuracy of different feature combinations, and the results are shown in Fig. 6.

Decision-level fusion is a concept used in various fields, including signal processing, machine learning, and artificial intelligence. Its purpose is to combine the outputs or decisions from multiple sources or classifiers to make a final decision that is more accurate, robust, and reliable than the individual decisions from each source. As can be seen in Fig. 6, when a single feature is used, its classification effect is stable around 80%, and the evaluation effect is relatively average, but after fusing the two information at the decision

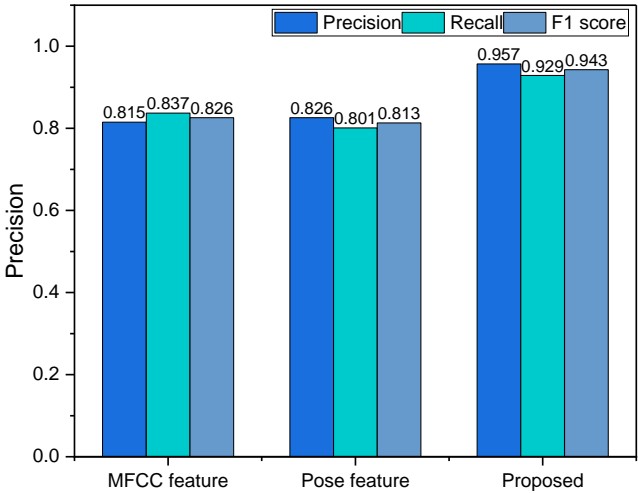

**Figure 6** **The evaluation result using different features.**

level its evaluation accuracy has improved substantially, which is very important for the future evaluation of music teaching effectiveness.

## Specific classification effects for different learning levels

To illustrate the effect of the model on the classification of students with different academic performance, we conducted a further analytical study of the data according to the three categories of "Excellent", "Good" and "Failed" at the beginning of the model design, the results of which are shown in Fig. 7.

The analysis of Fig. 7 offers valuable insights into the recognition results of the evaluation across students with different grades. The proposed method demonstrates superior recognition efficacy, particularly for students classified as "Excellent", wherein an impressive overall recognition rate of 96.7% is achieved. Despite some instances of misclassifications, the model's performance remains relatively balanced. This observation underscores the method's commendable ability to accurately identify and distinguish "Excellent" students based on their performance evaluation, thus showcasing its potential as an effective tool for educational assessment.

In conclusion, the analysis of Fig. 7 highlights the efficacy of the proposed method in recognizing students' performance across different grade categories, with a particularly noteworthy performance for "Excellent" students. While some misclassifications exist, the overall balance in the model's recognition results and its ability to distinguish students' academic achievements underscore its potential utility in educational assessment. The observation of a lower recognition rate for "Good" students is attributed to the inherent challenges in classifying students whose performance resides near grade boundaries, necessitating further investigation and potential enhancements to improve accuracy and mitigate bias. These findings contribute to the growing body of knowledge in educational

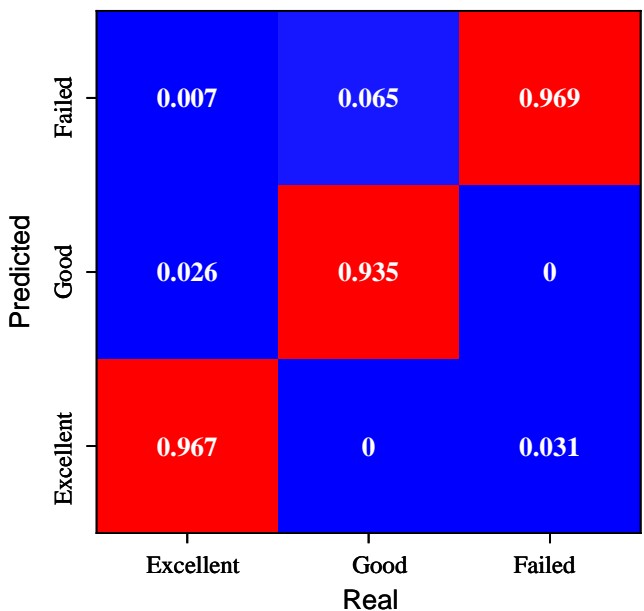

**Figure 7** The evaluation result for the students with different performance.

evaluation methodologies and hold promise for practical applications in real-world educational settings.

## DISCUSSION

This article completes the research of multimodal evaluation in the process of music teaching in universities using deep artificial neural network method, considering that the process of student performance evaluation needs not only to analyze the effect of their performance repertoire, but also to evaluate their related movements, so as to ensure the correctness of their subsequent learning. MFCC stands as a prominent frequency domain feature analysis method within the realm of speech recognition. This technique adeptly captures and preprocesses the characteristics of each frequency band. Upon applying convolution based on the CNN method to these features, we obtain convolution features, which facilitate subsequent processing. The efficacy of this approach significantly enhances processing efficiency and serves as a solid foundation for model training. Simultaneously, in the process of extracting human motion information, we employ OpenPose, an open-source framework, which proficiently computes and analyzes joint motion data of the human body. During the examination process, due to the relatively fixed environment, the extraction of students' motion feature information can be accomplished through video data acquisition. By amalgamating these two modalities, the evaluation performance of students with varying grades experiences a notable enhancement. This novel fusion of multimodal data paves the way for fresh insights into future applications.

In today's deep integration of IoT technology artificial intelligence, music education in schools should also introduce more intelligent ideas to improve teaching. In the

music teaching in colleges and universities, the demonstration of comprehensive musical ability should be strengthened more and the practice of comprehensive ability should be emphasized. Single skill training is the basis for improving comprehensive musical ability, but we should not pursue single skill too much, we need to set the right position of both, and we should pay attention to the training and cultivation of basic skills, but also emphasize the improvement of comprehensive ability (*Rolandson & Ross-Hekkel, 2022*). Secondly, the curriculum should focus on the development of students' comprehensive abilities, pay attention to reducing the number of courses, increase the content connection between disciplines, and avoid repetition. Students can be invited to appreciate and appreciate the sound material in combination with the audio material, avoiding the repetition of excellent repertoire. For example, improvisation accompaniment and self-playing singing are closely related to each other and both require the skills of improvisation accompaniment, which can be appropriately integrated together to reduce the burden of students' learning (*Wang, Ye & Yu, 2022*). Therefore, it is not only the use of intelligent methods in the evaluation process, but also the optimization of the curriculum and the enhancement of students' learning experience that is the optimal path to perfect talent development.

## CONCLUSION

In this article, a learning effect evaluation method based on multimodal fusion of sound information and action information is proposed for the evaluation of student learning performance in music teaching. Its main innovation lies in the completion of multimodal feature fusion between audio data and human motion data. During the feature layer fusion process, the accuracy of student learning effect evaluation is notably enhanced by combining sound information extracted from both the MFCC and CNN networks and action information derived from the deep network OpenPose framework. When these information sources were evaluated independently, the accuracy rate slightly exceeded 80%. However, upon completing the feature layer fusion, the recognition rate surged to over 95%. Subsequent tests on three distinct student groups consistently achieved recognition rates exceeding 90% after feature fusion for each group. These results underscore the substantial improvement in the accuracy of student learning effect evaluation through the fusion of multimodal features. Furthermore, this approach offers innovative insights for future music course evaluation and multimodal information fusion.

However, in the study, the amount of data selected was small, and in addition, no further discussion was conducted in the feature layer fusion process. Expanding the sample size and enriching the scenarios in future studies are both needed for further improvement.

## ACKNOWLEDGEMENTS

I would like to thank the anonymous reviewers whose comments and suggestions helped improve this manuscript.

### Funding

The author received no specific funding for this study.

### Competing Interests

The authors declare there are no competing interests.

### Author Contributions

- Yuexia Gao conceived and designed the experiments, performed the experiments, analyzed the data, performed the computation work, prepared figures and/or tables, authored or reviewed drafts of the article, and approved the final draft.

### Data Availability

The code is available in the Supplementary File and the data is available at Zenodo: Sebastia Vicenc Amengual Gari, Banu Sahin, Dusty Eddy, & Malte Kob. (2020). Open Database of Spatial Room Impulse Responses at Detmold University of Music (v0.2) [Data set]. 149th AES Convention. Zenodo. https://doi.org/10.5281/zenodo.4116247.

### Supplemental Information

Supplemental information for this article can be found online at http://dx.doi.org/10.7717/peerj-cs.1602#supplemental-information.

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
