# Peer review of "Application of multimodal perception scenario construction based on IoT technology in university music teaching"

_PeerJ Computer Science, doi:10.7717/peerj-cs.1602_

## Round 0.1 · original submission · Major Revisions

Authors must address the comments provided by reviewers.
Major review is required due to many issues highlighted by reviewers and I do agree with all the comments that have been provided.

**Language Note:** The review process has identified that the English language must be improved. PeerJ can provide language editing services - please contact us at [email protected] for pricing (be sure to provide your manuscript number and title). Alternatively, you should make your own arrangements to improve the language quality and provide details in your response letter. – PeerJ Staff

Reviewer 1 ·

Basic reporting

To improve the training level and accelerate the process of talent growth through intelligent means is the key direction of education development today. Aiming at the problem of evaluating students' learning status in college music teaching, a multi-modal information fusion based on sound information and motion information is proposed. The following inputs might help the authors to improve the paper.
1. It is suggested that the author re-select the keywords of the abstract according to the content of the paper;
2. There is a repetition at the end of the first section, please delete it;
3. After completing MFCC feature extraction, what is the function of using CNN method for further feature extraction in this paper;
4. The second paragraph on page 9 is misstated. It should be Figure 3 instead of Figure 2;
5. The loss function curve in Fig.4 is not compared with other loss functions and lacks experimental data support;
6. The explanation in Figure 6 mentions decision-level fusion. What is the role and purpose of this fusion?
7. What are the advantages of the Openpose framework, which uses CNN convolution, why not just use convolutional neural networks?
8. And if necessary, should add the process or method of data preprocessing, which will help other scholars to reference and learn from this study;
9. The innovation of the method proposed by the author is not strong, please highlight the innovation of the model;
10. Recent references should be added that it is better to include some references from good journal papers.

Experimental design

As commented in Basic Reporting.

Validity of the findings

As commented in Basic Reporting.

Additional comments

As commented in Basic Reporting.

Reviewer 2 ·

Basic reporting

No comment

Experimental design

No comment

Validity of the findings

No comment

Additional comments

In the training of instrumental talents, in addition to listening to their performance, it is also necessary to evaluate their movements to provide more information for the subsequent training. Aiming at the problem of evaluating students' learning status in music teaching in colleges and universities, a multi-modal information fusion evaluation method based on sound information and motion information is proposed, which provides a new idea for music teaching and interactive classroom teaching evaluation in the future, but there are still some deficiencies in this paper:
1. There are some repetitions in the last paragraph of the second page of the paper, the author carefully checked and revised;
2. Figure 1 lacks a corresponding explanation and display in the content of the paper;
3. In related works, some introductions can be added to the application of artificial intelligence in music teaching;
4. What features of sound are extracted by MFCC in this paper?
5. The formulas (9), (10) and (11) are the calculation of loss, but there is no specific explanation in the paper, and I will suggest the author to make a supplement;
6. I would suggest the author to add ablation experiments in the experimental chapter to enhance the feasibility of the experiment;
7. The confusion matrix data in Figure 7 seems to be wrong. The sum of the recognition rates of the horizontal and vertical axes equals 1. I would suggest the author to check and modify it;
8. The conclusions research results are similar to the previous contributions, and I will suggest the author to check and revise them;
9. The author needs to check the English grammar carefully to be ready for publication.

---

## Round 0.2 · accepted · Accept

The authors have addressed all the comments provided by the reviewers.

Reviewer 1 ·

Basic reporting

Improved

Experimental design

Improved

Validity of the findings

Improved

Additional comments

The manuscript is improved as suggested and I have no further concerns. I would like to accept the paper for publication.

Reviewer 2 ·

Basic reporting

No comment

Experimental design

No comment

Validity of the findings

No comment

Additional comments

The authors have satisfactorily resolved all of my concerns. Therefore, I suggest this article for publication in its current form.